# Anti-Virulence Activity of the Cell-Free Supernatant of the Antarctic Bacterium *Psychrobacter* sp. TAE2020 against *Pseudomonas aeruginosa* Clinical Isolates from Cystic Fibrosis Patients

**DOI:** 10.3390/antibiotics10080944

**Published:** 2021-08-04

**Authors:** Rosanna Papa, Gianluca Vrenna, Caterina D’Angelo, Angela Casillo, Michela Relucenti, Orlando Donfrancesco, Maria Michela Corsaro, Ersilia Vita Fiscarelli, Vanessa Tuccio Guarna Assanti, Maria Luisa Tutino, Ermenegilda Parrilli, Marco Artini, Laura Selan

**Affiliations:** 1Department of Public Health and Infectious Diseases, Sapienza University, p.le Aldo Moro 5, 00185 Rome, Italy; gianluca.vrenna@uniroma1.it (G.V.); laura.selan@uniroma1.it (L.S.); 2Department of Chemical Sciences, Federico II University, Complesso Universitario Monte S. Angelo, Via Cintia 4, 80126 Naples, Italy; caterina.dangelo@unina.it (C.D.); angela.casillo@unina.it (A.C.); corsaro@unina.it (M.M.C.); tutino@unina.it (M.L.T.); erparril@unina.it (E.P.); 3Department of Anatomy, Histology, Forensic Medicine and Orthopaedics, Sapienza University of Rome, Via Alfonso Borelli 50, 00161 Rome, Italy; michela.relucenti@uniroma1.it (M.R.); orlando.donfrancesco@uniroma1.it (O.D.); 4Unit Cystic Fibrosis Diagnostic Microbiology and Immunology Diagnostics, Diagnostic Medicine and Laboratory Department, Bambino Gesù Children’s IRCCS Hospital, 00165 Rome, Italy; evita.fiscarelli@opbg.net (E.V.F.); vanessa.tuccio@opbg.net (V.T.G.A.)

**Keywords:** *Pseudomonas aeruginosa*, cystic fibrosis, anti-virulence, SEM, motility, biofilm, pyocyanin, proteases, Antarctic bacteria

## Abstract

*Pseudomonas aeruginosa* is an opportunistic pathogen often involved in airway infections of cystic fibrosis (CF) patients. Its pathogenicity is related to several virulence factors, such as biofilm formation, motility and production of toxins and proteases. The expression of these virulence factors is controlled by quorum sensing (QS). Thus, QS inhibition is considered a novel strategy for the development of antipathogenic compounds acting on specific bacterial virulence programs without affecting bacterial vitality. In this context, cold-adapted marine bacteria living in polar regions represent an untapped reservoir of biodiversity endowed with an interesting chemical repertoire. In this paper, we investigated the biological activity of a supernatant derived from a novel Antarctic bacterium (SN_TAE2020) against specific virulence factors produced by *P. aeruginosa* strains isolated from FC patients. Our results clearly show a reduction in pyocyanin and protease production in the presence of SN_TAE2020. Finally, SN_TAE2020 was also able to strongly affect swarming and swimming motility for almost all tested strains. Furthermore, the effect of SN_TAE2020 was investigated on biofilm growth and texture, captured by SEM analysis. In consideration of the novel results obtained on clinical strains, polar bacteria might represent potential candidates for the discovery of new compounds limiting *P. aeruginosa* virulence in CF patients.

## 1. Introduction

Cystic fibrosis (CF) is a progressive genetic disease caused by the presence of mutations in the *cftr* gene, encoding for a protein called cystic fibrosis transmembrane conductance regulator (CFTR). This mutation has been reported to alter the transport across the cellular membrane, especially in the airways [1]. A defective CFTR protein produces thick, sticky mucus that clogs the airways, trapping microorganisms, leading to inflammation and recurrent and chronic infections, which cause respiratory failure and other complications [2].

Starting from childhood, *Staphylococcus aureus* (*S. aureus*) is one of the first colonizers of the CF airways, followed by *Haemophilus influenzae* (*H. influenzae)* and then, during adolescence and for all patient life, the opportunistic pathogen *Pseudomonas aeruginosa* (*P. aeruginosa*) [3,4].

*P. aeruginosa* is an aerobe and facultative anaerobe that can survive well in microaerobic conditions such as those generated by the sticky mucus within the lungs of CF patients [5,6].

The Infectious Diseases Society of America (IDSA) recently focused attention on clinical microorganisms—acronymically termed ‘ESKAPE pathogens’—capable of ‘escaping’ the antimicrobial action of antibiotics, representing new patterns in pathogenesis, transmission and resistance. ESKAPE microorganisms include both *S. aureus* and *P. aeruginosa*. A deep understanding of virulence mechanisms adopted by these bacteria during transmission and pathogenicity may lead to innovative strategies for the development of new anti-infective options [7].

During CF infection, the lung environment is extremely hostile because of the very high concentrations of antibiotics, reduced nutrient availability, elevated osmotic stress and intermicrobial competition: these conditions force *P. aeruginosa* to adapt for survival [8].

The virulence of each *P. aeruginosa* isolate is strongly related to different factors such as the capability to form a biofilm, different types of cell/colonial motility and production of toxins (for example, bacterial pigments) [9]. The dynamic process of biofilm formation offers protection to bacterial cells and resistance to drugs and host immune attacks. The self-produced exopolysaccharide matrix (EPS) that can incorporate different bacterial communities ensures their survival and resistance to certain antibiotics, complicating bacterial eradication [10]. Motility also contributes to biofilm formation and bacterial colonization of surfaces [11]. Swarming, swimming and twitching are three different types of *P. aeruginosa* motilities [12].

Other virulence factors are important for *P. aeruginosa* survival and persistence in a human host. Among these, pyocyanin production induces oxidative stress and is important for the establishment of infections, interfering in numerous cellular processes of the host [13].

The expression of these virulence factors is mainly controlled by quorum sensing (QS), an intercellular communication system that bacteria use to monitor population density via signaling molecules and receptors. This program self-adapts gene expression to the constantly changing needs [14,15]. Therefore, the control of QS may become a drug target for the development of new antipathogenic agents [16]. Quorum sensing inhibitors (QSIs) act on specific virulence programs and do not affect bacterial vitality. For this reason, they should not lead to the development of the resistance observed for antibiotics [17].

Recently, several agents of natural origin isolated from fungi [18,19], bacteria [20] and plants [21,22] have shown anti-virulence features against some bacterial species, including *P. aeruginosa*.

In this context, we chose to explore an exotic and unusual ecological niche represented by the polar marine habitat. The marine environment holds great potential as a possible source of novel drugs. Marine microorganisms possess a vast diversity of metabolic capabilities, since their ecological niche forces them to abruptly adapt to the diverse range of physical and chemical conditions of rapidly changing marine ecosystems. Biologically active compounds isolated from marine sources have already been selected for the treatment of various diseases, such as anticancer, antimicrobial and antifouling compounds [23,24]. Cold-adapted marine bacteria deriving from polar regions represent an unexploited reservoir of biodiversity endowed with an interesting chemical repertoire, also applicable in the pharmaceutical field [25].

*Psychrobacter* sp. TAE2020 is an aerobic ƴ-proteobacterium isolated from an Antarctic coastal seawater sample collected in the vicinity of the French Antarctic station Dumont d’Urville. Preliminary data indicated its ability to produce anti-biofilm molecules active on *Staphylococcus epidermidis* (submitted); therefore, this observation prompted us to evaluate the effect of molecules produced by the Antarctic bacterium against *P. aeruginosa* cells. This work aims to investigate the capability of *Psychrobacter* sp. TAE2020 to produce and secrete molecules active against some virulence factors produced by *P. aeruginosa* isolated from CF patients, such as biofilm formation and accumulation, pyocyanin production and swimming and swarming motility.

## 2. Results

### 2.1. Phenotypic Characterization of Clinical and PA14 Strains

The selection of *P. aeruginosa* isolates was based on specific virulence factors, such as pyocyanin production, proteolytic activity, biofilm formation and swarming and swimming motility. The characterization of these features in the selected strains is reported in Table 1.

### 2.2. Effect of SN_TAE2020 on Biofilm Formation

*Psychrobacter* sp. TAE2020 was grown in a synthetic medium named Gut in planktonic conditions at 15 °C until the stationary phase of growth (72 h). The supernatant (SN_TAE2020) was recovered by centrifugation and sterilized by filtration, and then it was stored at 4 °C until use. The supernatant did not show antimicrobial activity at all tested dilutions (data not shown).

Preliminary experiments were carried out to assess the effect of SN_TAE2020 on the growth rate of *P. aeruginosa* bacterial strains. SN_TAE2020 was added to the growth medium at two different concentrations (diluted 1:2 and 1:5), and bacterial growth curves were monitored over 30 h. As a control, bacteria were also grown in the presence of 50% of the culture medium used for TAE2020 growth (Gut medium). As shown in Figure 1, SN_TAE2020 did not significantly affect the *P. aeruginosa* duplication rate for each tested strain at both used concentrations. Indeed, bacterial growth curves were nearly superimposable both in the presence and in the absence of SN_TAE2020.

The effect of SN_TAE2020 was investigated at two different time points during biofilm development; it was added to the medium at the beginning of the cultivation (0 h, pre-adhesion period), and after biofilm formation (24 h of bacterial culture). SN_TAE2020 was used at a dilution of 50% in BHI 2X (final concentration of broth 1X). As a control, bacteria were cultured in the presence of the Gut medium. The results on the anti-biofilm activity of SN_TAE2020 added at the beginning of the cultivation period (0 h) are presented in Figure 2A.

The results are expressed as the percentage of biofilm formed in the presence of SN_TAE2020 compared to the bacteria grown in the presence of 50% of the Gut medium.

SN_TAE2020 showed its activity on all the tested strains, where inhibition of biofilm formation variably ranged from about 90% to 50%. The strongest inhibition (86.5%) was achieved on the isolate 23P. Furthermore, SN_TAE2020 also showed inhibitory activity on the highly virulent reference strain PA14, characterized by a specific mutation in the *retS* gene responsible for hyperbiofilm production [26].

SN_TAE2020 was also tested for its ability to degrade mature biofilms. In this case, bacterial isolates were grown in static conditions in microtiter plates for 24 h at 37 °C, and then the growth medium was replaced by adding SN_TAE2020 at 50% dilution. As a control, the growth medium supplemented with Gut broth at a dilution of 1:2 was added in the corresponding wells. The microtiter plates were incubated for a further 24 h at 37 °C. The results are presented in Figure 2B. After the addition of SN_TAE2020, in two out of six strains, a biofilm reduction was observed, ranging from 40% to 60%. However, it is worth noting that biofilms measured after 48 h of incubation with medium replacement after 24 h were more abundant and probably more structured and difficult to eradicate. Considering the difficult eradication, a higher concentration of SN_TAE2020 could be necessary to observe a more marked eradication effect.

### 2.3. Effect of SN_TAE2020 on Pyocyanin Production and Protease Activity

SN_TAE2020’s influence on pyocyanin production and protease activity was assessed on all tested *P. aeruginosa* strains which had been chosen for their strong production of these virulence factors (Table 1).

First, we investigated the effects of SN_TAE2020 diluted 1:5 in culture medium on the synthesis and secretion of the virulence-associated pigment pyocyanin. As a control, bacteria were cultured in the presence of 20% (1:5 dilution) of the same broth used for TAE2020 growth (Gut medium). For each strain, data are expressed in Figure 3 as a percentage of residual pyocyanin production after 48 h of treatment in comparison with that of the control experiment. Pyocyanin production was also normalized to the optical density reached by the bacterial strain at 48 h.

SN_TAE2020 reduced pyocyanin production in five out of six tested strains. On the reference strain PA14 and on the clinical isolate 27P, the reduction in pyocyanin production was higher than 50%.

Proteases, in particular, LasA and LasB of *P. aeruginosa*, are considered as major virulence factors playing a pivotal role during acute and chronic infections [27]. In the present study, the ability of SN_TAE2020 diluted 1:5 in culture medium to reduce azocasein-degrading protease activity was observed. The residual proteolytic activity in the culture supernatant of *P. aeruginosa* was compared with that of untreated bacteria after 48 h of growth. As a control, bacteria were cultured in the presence of 20% (1:5 dilution) of the same broth used for TAE2020 growth. The results, expressed as a percentage of residual proteolytic activity of the treated cultures in comparison with that of the corresponding controls (growth in the presence of Gut diluted 1:5), are reported in Figure 4. Proteolytic activity was also normalized to the optical density reached by the bacterial strain at 48 h. A partial reduction in proteases was highlighted in three out of six tested strains.

### 2.4. Effect of SN_TAE2020 on Motility

The ability of SN_TAE2020 to reduce swarming and swimming motility patterns in reference and clinical *P. aeruginosa* isolates was also investigated. The results are represented in Figure 5.

Swarming is a mode of surface translocation dependent on cell-to-cell signaling that requires flagella and pili. SN_TAE2020 was diluted 1:5 in the appropriate semi-solid medium, and, as a control, a 1:5 dilution of the growth medium used for the TAE2020 culture was used. As reported in Figure 5, SN_TAE2020 reduced swarming zones in all tested strains.

In the soft agar plate of the treated *P. aeruginosa* 31P, bacteria were not macroscopically visible despite the concentration of SN_TAE2020 used being unable to impair bacterial vitality.

Regarding swimming motility, produced by polar flagella movements of bacteria in a liquid environment, the results are reported in the right panel of Figure 5. In all tested strains, SN_TAE2020 caused a strong reduction in swimming motility. *P. aeruginosa* 31P was not macroscopically visible on the soft agar plate in this case too, despite the adopted concentration of SN_TAE2020 being unable to impair bacterial vitality.

### 2.5. Morphological Evaluation by SEM Analysis

#### 2.5.1. Evaluation of SN_TAE2020’s Influence on *P. aeruginosa* PA14

The untreated *P. aeruginosa* PA14 biofilm sample exhibits a smooth, compact and dense aspect (Figure 6A), and a few micro-channels are visible. At a higher magnification (Figure 6B), one bacterial cell is observed on the matrix surface. The cell surface is rough due to the presence of pili wrapped around the cell body. When *P. aeruginosa* PA14 is treated with SN_TAE2020, the biofilm topography changes, the dense and compact appearance is modified into a more porous one and the biofilm is partially disaggregated due to the irregularly sized pores that perforate its surface (Figure 6C). At higher magnifications, bacterial cells are observed both on the biofilm surface or detached from the spongy biofilm matrix (Figure 6D).

#### 2.5.2. Evaluation of SN_TAE2020 on *P. aeruginosa* 23P

The untreated *P. aeruginosa* 23P biofilm control sample shows a rough and dense surface aspect (Figure 7A), and a few deep micro-channels, with an uneven diameter, are visible. At a higher magnification (Figure 7B), some bacterial cells are observed emerging from the matrix surface. When *P. aeruginosa* 23P is treated with SN_TAE2020, the biofilm topography changes, very large ladders characterize the biofilm surface (Figure 7C) and the dense matrix is modified into a spongy and meshed structure. At higher magnifications, the biofilm appears to be formed by a loose and stratified network of extracellular polymeric substances in which bacterial cells are more or less entrapped (Figure 7D).

### 2.6. Anti-Virulence Features of Organic Extract from the Supernatant of TAE2020

The cell-fee supernatant was extracted with an organic solvent, as described in the Materials and Methods section, to generate EX_TAE2020 and analyzed for its anti-virulence features. The results obtained are summarized in Figure 8. It was tested on the reference strain *P. aeruginosa* PA14 at a concentration of 2 mg/mL, as previously reported [28], to investigate its capability to impair biofilm formation, protease activity, swarming motility and pyocyanin production. First, we tested EX_TAE2020 as an antimicrobial, and the results show that it did not affect bacterial vitality at all tested concentrations (data not shown).

Despite the minor complexity of the extract, it showed the ability to preserve several of the anti-virulence capabilities previously highlighted by the supernatant. However, EX_TAE2020, contrary to SN_TAE2020, was not able to reduce pyocyanin production.

### 2.7. GC-MS Analysis of the Organic Extract from SN_TAE2020

The organic extract from SN_TAE2020 was subjected to chemical analyses to detect the presence of fatty acids, carbohydrates and amino acids. The GC-MS chromatogram of the glycosyl analysis did not reveal any monosaccharides, whereas the amino acids glutamic acid, phenyl alanine and leucine were detected. From the fatty acid analysis, C14:0 to C18:0, C16:1 and C18:1 were detected. In agreement with these data, the ^1^H-NMR (Figure 9) and the MALDI-TOF (data not shown) spectra suggested a mixture of low-molecular mass molecules. The signals at low values of proton chemical shifts, in the range from 0.5 to 2 ppm, can be attributable to the methylene (CH_2_) and methyl groups (CH_3_) of long hydrocarbon chains, confirming the presence of fatty acids. The signals between 2.0 and 6.0 ppm can be attributable to the residues of amino acids, as well as the signals around 8 ppm. The positive ion MALDI-TOF spectrum confirmed that the fraction was constituted by a complex mixture of low-molecular mass compounds since it was rich in signals between 600 and 1600 Da.

## 3. Discussion

*P. aeruginosa* possesses an inherent resistance to a wide variety of antimicrobials due to its low membrane permeability, expression of different efflux pumps and biofilm production that reduce the physical access of immune effectors and therapeutics, reducing the success of antibacterial treatment. Moreover, mainly during chronic infections, such as those occurring in the lungs of CF patients, *P. aeruginosa* can easily acquire new resistance mechanisms via horizontal gene transfer [24].

Taking into account these assumptions and in consideration of the growing threat of antimicrobial resistance, it is mandatory to focus research not only on the discovery of new antimicrobials but also on the development of innovative therapeutic treatments. A promising strategy could be based on targeting the virulence features of a pathogen, in order to block its ability to inflict injury, helping to slow down pathogenetic mechanisms and restore bacterial clearance by the host [29,30].

Colonization of host tissues, implants and medical devices and environmental surfaces facilitated by the formation of biofilms is often the basis for the development of chronic infection [31]. The development of a biofilm is a multistep process that requires different actors, including various motility factors, followed by the secretion of components of the extracellular matrix, all finely regulated by the hierarchical QS system [32].

Due to their relevance, QS and biofilms are the most frequently targeted systems to disarm *P. aeruginosa* virulence [29,33].

Furthermore, QS systems in *P. aeruginosa* also control the expression of several other virulence factors, including proteases, exotoxin A, pyocyanin, pili and flagella related to bacterial motility [34].

The literature reports several compounds of synthetic or natural origin that inhibit (defined as QSI) or quench QS-dependent gene expression. Several studies report the identification of various QSIs and quorum quenchers that attenuate the production of virulence factors in vitro and moderate *P. aeruginosa* virulence without exhibiting antimicrobial activity. They also increase bacterial susceptibility to antibiotics by inhibiting biofilm formation.

In this report, we investigated the anti-virulence properties of a supernatant derived from an Antarctic bacterium named *Psychrobacter* sp. TAE2020 against specific virulence factors produced by *P. aeruginosa* isolates obtained from FC patients. Indeed, the investigation of new unexplored habitats and uncommon environments has become an important source for the discovery of novel bacterial metabolites with antimicrobial and anti-virulence activity.

Our previously reported studies showed antimicrobial and anti-biofilm activities expressed by different marine cold-adapted bacteria isolated from polar regions, especially Antarctica, against staphylococci and *P. aeruginosa* [20,28,35,36,37]. Herein, we demonstrated that the cell-free supernatant of an Antarctic bacterium belonging to the *Psychrobacter* genus, named TAE2020, strongly affected some specific virulence features of *P. aeruginosa* isolates from CF patients, without affecting bacterial vitality or inhibiting bacterial growth.

Regarding biofilm inhibition, SN_TAE2020 strongly affected the biofilm development of all tested strains with different efficacies. In particular, the effect seemed to be more marked for stronger biofilm producers (strains PA14, 23P and 31P). Particularly interesting is the disaggregating activity showed by SN_TAE2020 on mature biofilms, since once a biofilm is established, it is very hard to disrupt it. We observed a positive effect on biofilm disaggregation in two out six of the bacterial strains analyzed, with an efficiency of about 50% (Figure 1).

Morphological analysis of the biofilm obtained by SEM returned interesting information on the action of SN_TAE2020 on a *P. aeruginosa* reference strain (PA14) and on the clinical *P. aeruginosa* strain 23P. The latter strain was chosen because it was a strong biofilm former, and its biofilm growth was strongly impaired by SN_TAE2020. Major structural differences exist in the ECM structure of the two untreated strains. In fact, the two control samples showed different biofilm features, which were more compact for PA14 compared to 23P. On both samples, the effect of the treatment was clearly visible, but it was more pronounced on 23P, probably because the structure of the untreated biofilm built by this strain was less dense. SEM images showed that the SN_TAE2020 treatment detaches and disperses PA14 bacterial cells from the polymeric matrix of the biofilm; this could suggest a weakening of the bonds between bacteria and matrix components. This does not happen in sample 23P, where bacteria are still mostly immersed in the matrix, while the matrix itself is ‘unstranded’; in this case, the mechanism of action is probably different and concerns only the matrix.

In both cases, we can explain why the motility of bacteria decreases; in sample PA14, the activity of pili and flagella is probably involved, while in 23P, the disrupted matrix probably entangles the bacteria and holds them back, preventing their movement.

Besides its biofilm, *P. aeruginosa* possesses other virulence features. Pyocyanin, for example, interferes with numerous functions in host cells due to its high redox activity and is used by the bacterium to establish infections and to promote virulence [19]. The production and secretion of this green pigment are controlled by QS and allow bacterial persistence in hostile environments, by aiding biofilm development [38]. Pyocyanin also exerts an antimicrobial activity against several bacteria and fungi and favors lung damage and the consequent respiratory failure by an increase in oxidative stress in the lung epithelium [39]. Our results show a reduction in pyocyanin production in five out of six tested strains (with an efficacy higher than 50% for two strains), likely suggesting a possible interference with QS activation and regulation.

A partial reduction in proteases was highlighted in three out of six tested strains after treatment with SN_TAE2020; this is an extremely interesting feature of the tested supernatant, since the proteolytic activity of *P. aeruginosa*, particularly due to extracellular proteases LasA and LasB, is considered a major virulence factor playing a pivotal role during acute and chronic infections [27]. However, it is interesting to note that in these experiments, the supernatant was used at a lower concentration than that adopted for the adhesion experiments, since the high salt concentration of the medium used for *Psychrobacter* growth inhibits the production of *P. aeruginosa* pigment and proteases. In order to exclude an effect due to the medium salt concentration and to render the sample less complex, the supernatant was extracted with an organic solvent.

A stronger inhibition of the proteolytic activity was obtained using the organic extract derived from the supernatant, which allowed the use of higher concentrations since the experiment did not depend on the growth medium.

Unfortunately, the extract did not induce a reduction in pyocyanin production. However, it is well known that the production of pyocyanin and other virulence factors is controlled by different regulators within the QS system [13]; therefore, the activities performed by the supernatant are not necessarily due to a single compound acting on all the analyzed virulence factors.

Finally, SN_TAE2020 was able to strongly affect *P. aeruginosa* swarming and swimming motility for almost all tested strains. Additionally, in this case, inhibition of motility (evaluated on swarming motility) was confirmed for the extract.

Preliminary characterization of the organic extract was carried out by means of chemical analyses and spectroscopic techniques. The amino acid content, analyzed by GC-MS as N-acetyl methyl ester derivatives, revealed the presence of glutamic acid, phenylalanine and leucine. Furthermore, the GC-MS analysis of fatty acids derivatized as fatty acid methyl ester suggested the presence of C14:0 to C18:0, C16:1 and C18:1. Finally, the ^1^H-NMR and the MALDI-TOF spectra suggested that the organic extract is a mixture of compounds. In consideration of the novel results obtained on clinical strains isolated from CF patients, the marine Antarctic bacterium *Psychrobacter* TAE2020 could be considered as a good candidate for the identification of anti-virulence compounds. These compounds could be candidates for the disinfection of the equipment adopted to improve the respiratory function of patients (masks, ventilators, etc.), since contaminated objects represent a route of indirect transmission of virulent strains [40]. Furthermore, several studies have reported promising synergy between antimicrobials and anti-virulence compounds. Many of these directly or indirectly target *P. aeruginosa* virulence features to increase bacterial susceptibility to antimicrobials.

## 4. Materials and Methods

### 4.1. Bacterial Strains and Growth Conditions

For this study, five *P. aeruginosa* strains were used, isolated from respiratory specimens of five CF patients in a follow-up at Pediatric Hospital Bambino Gesù (OPBG) of Rome, Italy. Phenotypic characteristics of the bacterial strains are summarized in Table 2.

*P. aeruginosa* PA14 was used as a reference strain for studies on pathogenesis and biofilm formation [26]. Bacteria were grown in Brain Heart Infusion broth (BHI, Oxoid, Basingstoke, UK). Planktonic cultures were grown in flasks under vigorous agitation (180 rpm), while biofilm formation was performed in static conditions, at 37 °C.

*Psychrobacter* sp. TAE2020 was grown in flasks in synthetic medium Gut (L-Glutamic acid, sodium salt 10 g/L, NaCl 10 g/L; NH_4_NO_3_ 1 g/L; KH_2_PO_4_∙7H_2_O 1 g/L; MgSO_4_∙7H_2_O 200 mg/L; FeSO_4_∙7H_2_O 5 mg/L; CaCl_2_∙2H_2_O 5 mg/L) in planktonic conditions at 15 °C under vigorous agitation (180 rpm) until the stationary phase of growth (72 h). The supernatant was recovered by centrifugation at 7000 rpm at 4 °C for 30 min, sterilized by filtration through membranes with a pore diameter of 0.22 μm and stored at 4 °C until use.

### 4.2. Pre-Adhesion Period

Biofilm production was quantified based on microtiter plate biofilm assay (MTP), as previously reported [42]. Briefly, the wells of a sterile 96-well flat-bottomed polystyrene plate were filled with BHI containing a 1:100 dilution of overnight bacterial cultures (about 0.5 OD 600 nm). As control, the first row contained the untreated bacterial cells in BHI broth supplemented with the medium used for the growth of TAE2020 diluted 1:2. In the second row, SN_TAE2020 was added to the same bacterial culture at a dilution of 1:2. The plates were aerobically incubated for 18 h at 37 °C. After the incubation, the well content was aspirated and washed three times with double-distilled water to remove planktonic cells, and the plates were dried in an inverted position. For the quantification of biofilm formation, each well was stained with 100 µL of 0.1% crystal violet, incubated for 15 min at room temperature, rinsed twice with double-distilled water and thoroughly dried. The remaining dye attached to the adherent cells was solubilized with 20% (*v/v*) glacial acetic acid and 80% (*v/v*) ethanol. After 30 min of incubation at room temperature, the total biofilm biomass in each well was spectrophotometrically quantified at 590 nm. Each data point is composed of 4 independent experiments, each performed in at least 6 replicates.

### 4.3. Mature Biofilm

Assays on preformed biofilms were also performed. The wells of a sterile 96-well flat-bottomed polystyrene plate were filled with 100 µL of BHI medium containing 1:100 dilution of overnight bacterial culture. The plates were aerobically incubated for 24 h at 37 °C. Then, the contents of the plates were poured off, and the wells were washed to remove the unattached bacteria; an amount of 100 μL of the fresh medium containing SN_TAE2020 diluted 1:2 was added into each well. As control, 100 μL of the fresh medium containing Gut diluted 1:2 was added. The inoculated plates prepared in this way were aerobically incubated for an additional 24 h (48 h in total) at 37 °C. After 24 h, the plates were analyzed as previously described.

### 4.4. Pyocyanin Assay

Pyocyanin production was determined as described by Pejčić et al. (2020) [9], with modifications. Bacterial cells were inoculated in BHI broth with or without SN_TAE2020 at a dilution of 1:5 and incubated for different times at 37 °C. As control, the BHI was supplemented with the medium used for the growth of TAE2020 diluted 1:5. The cells were removed by centrifugation (10,000 rpm, 15 min), and the supernatant was used for pyocyanin extraction. Briefly, 2 mL of chloroform was added into 2 mL of the supernatant. The solution was mixed for 2 min by inversion and then decanted for 15 min to allow the separation of the organic phase from the aqueous phase. The lower layer containing pyocyanin was transferred into a tube containing 2 mL of 0.2 M HCl. The resulting solution was mixed and decanted to allow the separation of the two phases. Then, the pink-colored upper layer was separated, and pyocyanin was subsequently spectrophotometrically quantified at 520 nm. Pyocyanin was normalized to the optical density reached by each bacterial culture.

### 4.5. Protease Assay

The total proteolytic activity of *P. aeruginosa* was determined by the azocasein assay. An amount of 150 μL of both extract-treated and untreated culture supernatants was added to 500 μL of 0.3% azocasein (Sigma, St. Louis, MO, USA) in 0.05 M Tris–HCl and 0.5 mM CaCl_2_ pH 7.5 and incubated at 37 °C for 30 min. In the untreated sample, bacteria were grown in BHI supplemented with the medium used for the growth of TAE2020 diluted 1:5. The enzymatic reaction was stopped by adding an equal volume of l0% ice-cold trichloroacetic acid and incubating the reaction at 4 °C for 10 min. After incubation, the insoluble azocasein was removed by centrifugation at 10,000 rpm for 10 min, and the obtained supernatant was measured at OD 400 nm. Proteolytic activity was normalized to the optical density reached by each bacterial culture.

### 4.6. Motility Assays

#### 4.6.1. Swarming Assay

The swarming assay was performed as previously described by Yang and coworkers (2018) [43], with some modifications. Briefly, SN_TAE2020 was added to molten swarming agar at a dilution of 1:5. Swarming agar was prepared as follows: 0.8% Nutrient Broth (Oxoid, Basingstoke, UK), 0.5% D-(+)-glucose (Sigma, Steinheim, Germany) and 0.5% agarose (Invitrogen, Paisley, UK). The culture was then dispensed onto Petri dishes after gentle mixing. Once the culture was solidified, 2 μL of each overnight *P. aeruginosa* culture was inoculated in the center of the agar and then incubated at 37 °C for 24 h. We used the medium used for the growth of TAE2020 as control (diluted 1:5). After the incubation period, the diameters of the growth zones were measured. Anti-QS properties were identified by the reduction in swarming motility.

#### 4.6.2. Swimming Assay

The swimming assay was conducted according to previous research [43], with some modifications. The procedures were the same as those of the swarming assay, except for the swimming agar composition, which consisted of 1.0% peptone (Oxoid, Basingstoke, UK), 0.5% sodium chloride (Sigma, Steinheim, Germany) and 0.3% Bacto-Agar (BD, Le Pont de Claix, France). After the incubation period, the diameters of the growth zones were measured.

### 4.7. SEM Analysis

For SEM analysis, bacteria were grown as follows: briefly, 1:100 dilution of overnight bacterial cultures was transferred to tubes containing SEM stubs (aluminum, 12.5 mm diameter, 6 mm pin) and incubated for 18 h at 37 °C in static conditions allowing biofilm production, in BHI and in the presence or absence of SN_TAE2020 diluted 1:2. After the growth, SEM stubs were washed in 0.1 M phosphate buffer pH 7.4 (PB) and fixed in 2.5% glutaraldehyde in 0.1 M PB buffer. Samples were washed overnight in PB and postfixed with a mixture of 2% OsO_4_ and 0.2% Ruthenium Red, for 1 h at room temperature [44,45,46]. Samples were then washed for 30 min with H_2_O, the excess of water was dried carefully with filter paper and then the samples were mounted onto the specimen holder. P14 was covered by Hitachi ionic liquid Hilem 1000, 10% in H_2_O, then carefully dried with filter paper and finally observed in a Hitachi SU3500 VP-SEM microscope (Hitachi, Tokyo, Japan), at an operating condition of 3 kV and high vacuum. 23P was observed in the same microscope but using variable pressure conditions of 5 kV and 30 Pa.

### 4.8. SN_TAE2020 Supernatant Preparation and Organic Extraction Protocol

The cell-free supernatant from *Psychrobacter* sp. TAE2020 culture was recovered by centrifugation at 7000 rpm at 4 °C for 30 min. After centrifugation, the supernatant was separated from the cells and sterilized by filtration through membranes with a pore diameter of 0.22 μm. The cell-free supernatant, named SN_TAE2020, was frozen at −20 °C until use. SN_TAE2020 was subjected to a liquid–liquid extraction to obtain the extracellular extract without adding cryoprotectants. In detail, it was thawed and stirred with ethyl acetate in a volume ratio of 2:1 (assay percent range ≥ 99.5%) (Sigma-Aldrich, St. Louis, MO, US) and mixed at 1% with formic acid (assay percent range = 90%; JT Baker, Munich, Germany). The solution was stirred for at least 30 min and subsequently centrifuged at 3000 rpm for 30 min. The resulting two phases were separated; the organic phase was recovered and dried by using a rotary evaporator, Rotavapor (Buchi R-210, Rodano (MI) Italy), at a temperature lower than 40 °C. The resulting organic extracts (EX_TAE2020) were aliquoted and stored at −20 °C.

### 4.9. GC-MS of SN_TAE2020

All the samples were analyzed with the GC-MS Agilent Technologies 7820A equipped with a mass selective detector, 5977B (HP-5 capillary column, 30 m × 0.25 mm i.d.; flow rate, 1 mL min^−1^, He as carrier gas). To a sample of the organic extract (1 mg), 6 M HCl (0.1 mL) was added, and the reaction was left at 110 °C for 16 h. After neutralization and drying, the sample was treated at 80 °C for 90 min with 0.5 M of HCl/CH_3_OH. After drying under a stream of air, the sample was acetylated as reported [47]. The N-acetyl methyl ester of amino acids obtained was injected adopting the following temperature program: 90 °C for 3 min, from 90 to 300 °C at 15 °C/min and at 300 °C for 5 min.

For the glycosyl and fatty acid analyses, another sample of the extract (1 mg) was treated with 0.5 mL of 1.25 M HCl/CH_3_OH, and the reaction was carried out at 80 °C for 16 h. After this time, the sample was extracted with hexane, and both the methanol and hexane layers were dried. The methanol extract was acetylated as already reported [47] prior to injection into the GC-MS, whereas the hexane fraction was analyzed after being dissolved in acetone. The temperature program for acetylated methyl glycosides was as follows: 140 °C for 3 min, 140→240 °C at 3 °C/min.

Finally, the temperature program for the fatty acid analysis was as follows: 140 °C for 3 min, then 140→280 °C at 10 °C/min and, finally, 280 °C for 20 min.

### 4.10. NMR Analysis of SN_TAE2020

An amount of 6 mg of the extract was dissolved in 0.5 mL CDCl_3_/CD_3_OD 1:1 and analyzed by NMR. The ^1^H NMR experiment was recorded at 298 K with a Bruker Avance 600 MHz equipped with a cryoprobe.

### 4.11. Mass Spectrometry Analysis of SN_TAE2020

The MALDI-TOF mass spectrum of the extract was obtained as already reported [48]. Briefly, 100 μg of the sample was dissolved in 0.1 mL CHCl_3_/CH_3_OH, and 0.5 μL was loaded on the MALDI target together with 0.5 μL of a matrix solution of 2,5-dihydroxybenzoic acid (DHB). The positive ion spectrum was acquired in reflectron mode on an ABSCIEX TOF/TOF 5800 (AB SCIEX, Darmstadt, Germany) mass spectrometer equipped with an Nd:YLF laser.

## 5. Conclusions

Compounds from Antarctic bacteria could be candidates for the disinfection of the equipment adopted by CF patients (masks, ventilators, etc.). Several studies have reported promising synergy between antimicrobials and anti-virulence compounds, those target *P. aeruginosa* virulence features to increase bacterial susceptibility to antimicrobials.

## Figures and Tables

**Figure 1 antibiotics-10-00944-f001:**
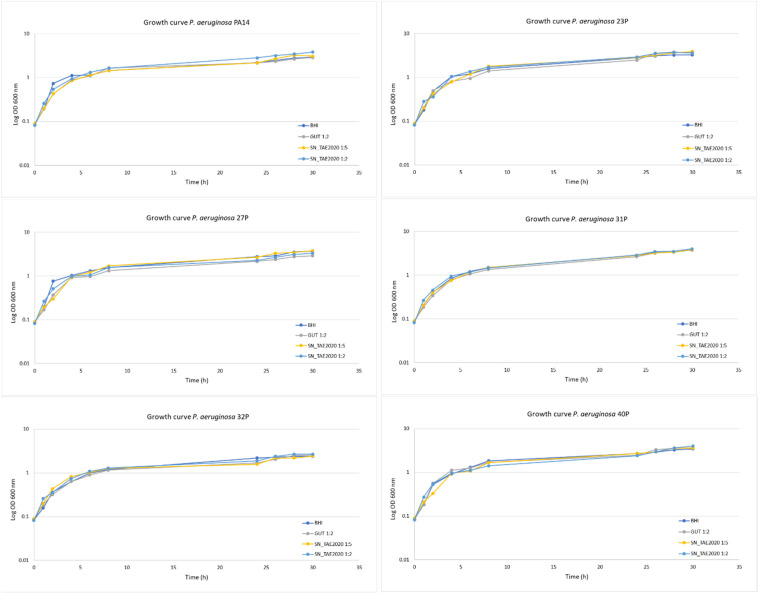
Growth curves of *P. aeruginosa* bacterial strains at 37 °C at 180 rpm. *P. aeruginosa* bacterial strains were grown in: BHI, in the presence of 20% and 50% of SN_TAE2020, and in the presence of Gut medium at 50%. Results are representative of three independent experiments.

**Figure 2 antibiotics-10-00944-f002:**
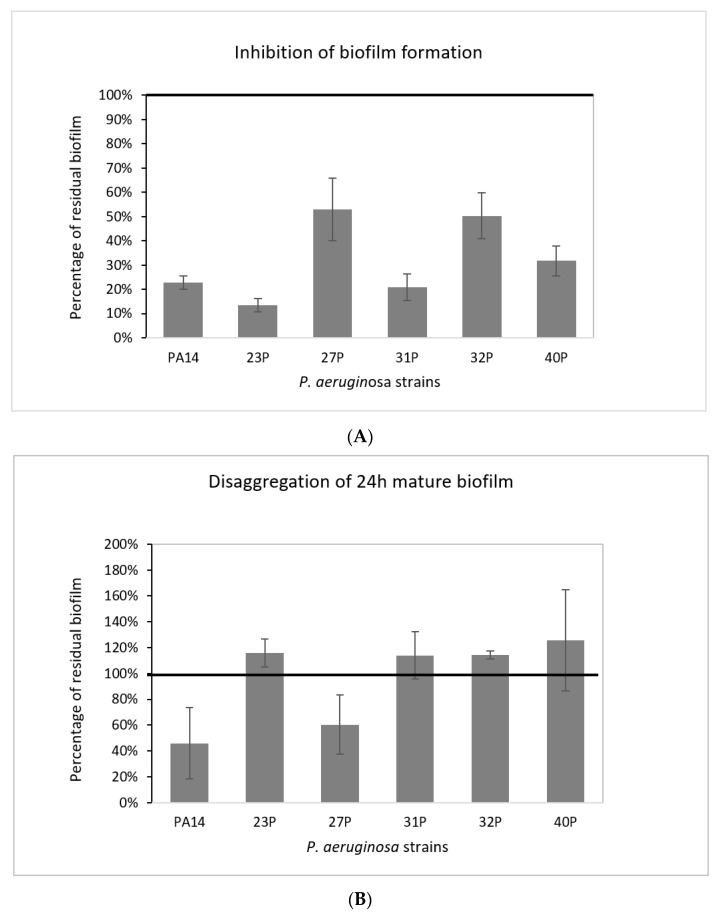
Effect of TAE2020 supernatant (SN_TAE2020) on biofilms of different clinical and reference strains. (**A**) Effect of SN_TAE2020 on biofilm formation. On the ordinate axis, the percentage of bacterial biofilm production is reported. Data are expressed as the percentage of biofilm formed in the presence of SN_TAE2020 compared with the control sample. Each data point is composed of 4 independent experiments, each performed in at least 3 replicates. (**B**) Effect of SN_TAE2020 on 24 h mature biofilm. On the ordinate axis, the percentage of residual biofilm is reported. Data are expressed as the percentage of residual biofilm after 24 h of treatment compared with the control sample. Each data point is composed of 4 independent experiments, each performed in at least 3 replicates.

**Figure 3 antibiotics-10-00944-f003:**
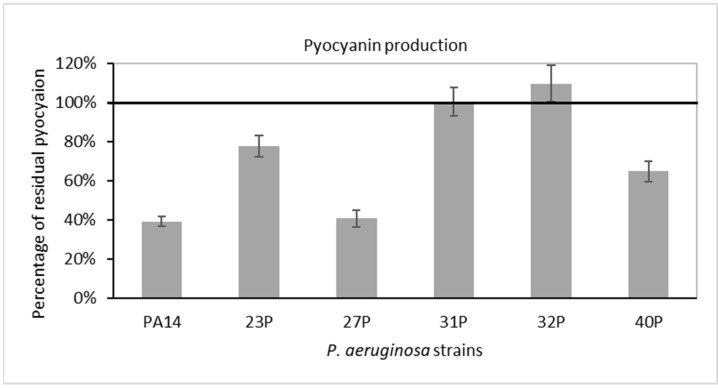
Effect of SN_TAE2020 on pyocyanin production after 48 h of incubation. Data are reported as a percentage of residual pyocyanin production after the SN_TAE2020 treatment in comparison with controls (bacteria grown in the presence of Gut diluted 1:5). Each data point is composed of 4 independent experiments, each performed in at least 3 replicates.

**Figure 4 antibiotics-10-00944-f004:**
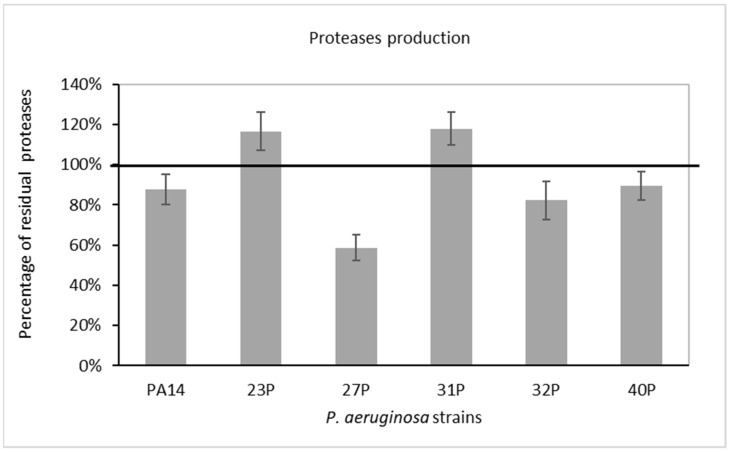
Effect of SN_TAE2020 on protease production after 48 h of incubation. Data are reported as the percentage of residual proteolytic activity after the SN_TAE2020 treatment in comparison with that of controls (bacteria grown in the presence of Gut diluted 1:5). Each data point is composed of 4 independent experiments, each performed in at least 3 replicates.

**Figure 5 antibiotics-10-00944-f005:**
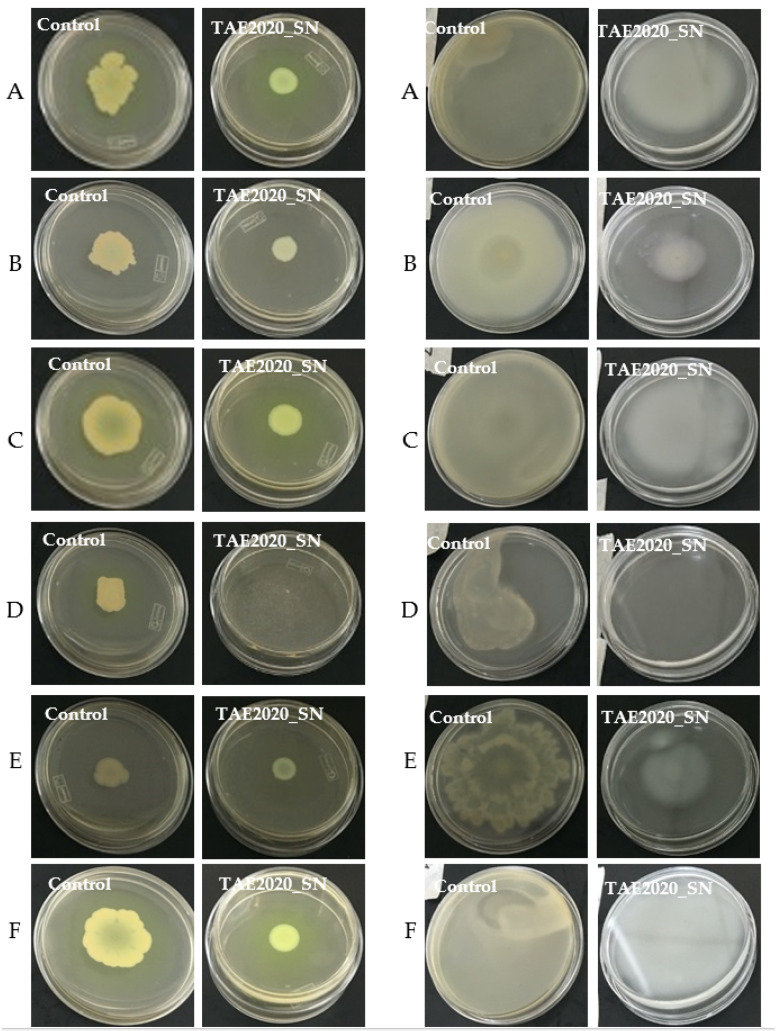
Motility assay. **Left panel:** Swarming inhibition assay of SN_TAE2020 on reference and clinical strains. (**A**) *P. aeruginosa* PA14; (**B**) *P. aeruginosa* 23P; (**C**) *P. aeruginosa* 27P; (**D**) *P. aeruginosa* 31P; (**E**) *P. aeruginosa* 32P; (**F**) *P. aeruginosa* 40P. **Right panel:** Swimming inhibition assay of SN_TAE2020 on reference and clinical strains. (**A**) *P. aeruginosa* PA14; (**B**) *P. aeruginosa* 23P; (**C**) *P. aeruginosa* 27P; (**D**) *P. aeruginosa* 31P; (**E**) *P. aeruginosa* 32P; (**F**) *P. aeruginosa* 40P.

**Figure 6 antibiotics-10-00944-f006:**
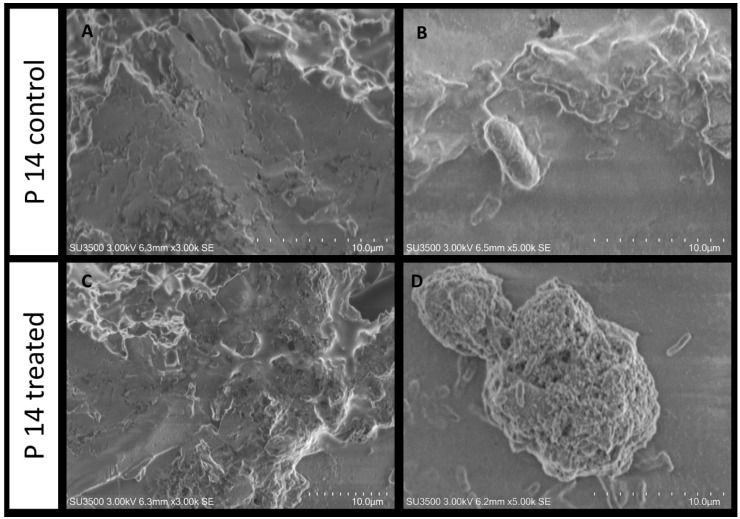
SEM analysis. *P. aeruginosa* PA14 untreated (**A**,**B**) and treated (**C**,**D**) with SN_TAE2020. SEM (**A**) 3000×, biofilm appears as a dense and smooth-surfaced mass; (**B**) 5000×, a bacterial cell is adherent to the matrix surface; (**C**) 3000×, the treatment affects the biofilm surface, and deep holes are seen partially perforating its surface; (**D**) 5000×, a globular mass of biofilm with a spongy aspect is shown, where bacterial cells are dispersed on its surface or lay detached and sparsely on the surroundings.

**Figure 7 antibiotics-10-00944-f007:**
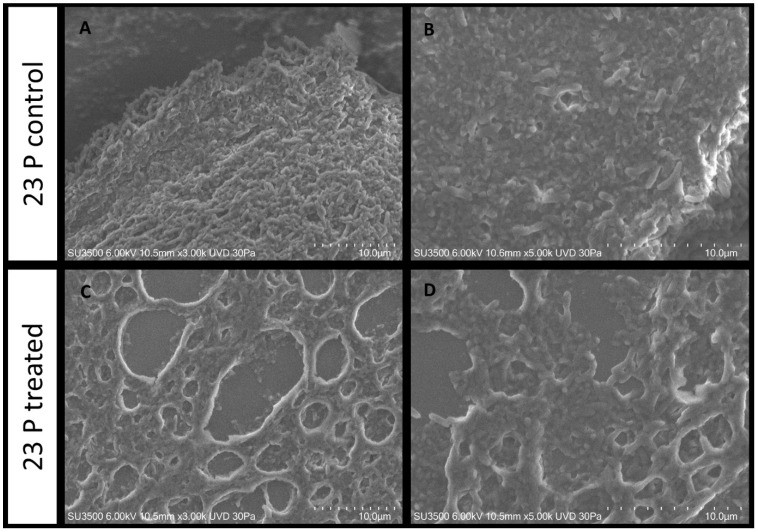
*P. aeruginosa* 23 P untreated (**A**,**B**) and treated (**C**,**D**) with SN_TAE2020.VP-SEM (**A**) 3000×, biofilm appears as a dense and rough-surfaced structure; (**B**) 5000×, bacterial cells are visible on the matrix surface and entrapped in the matrix bulk; (**C**) 3000×, the treatment affects the biofilm surface, and large ladders disrupt its surface; (**D**) 5000×, image at higher magnification shows an irregularly stratified network of extracellular polymeric substances in which bacterial cells are more or less entrapped.

**Figure 8 antibiotics-10-00944-f008:**
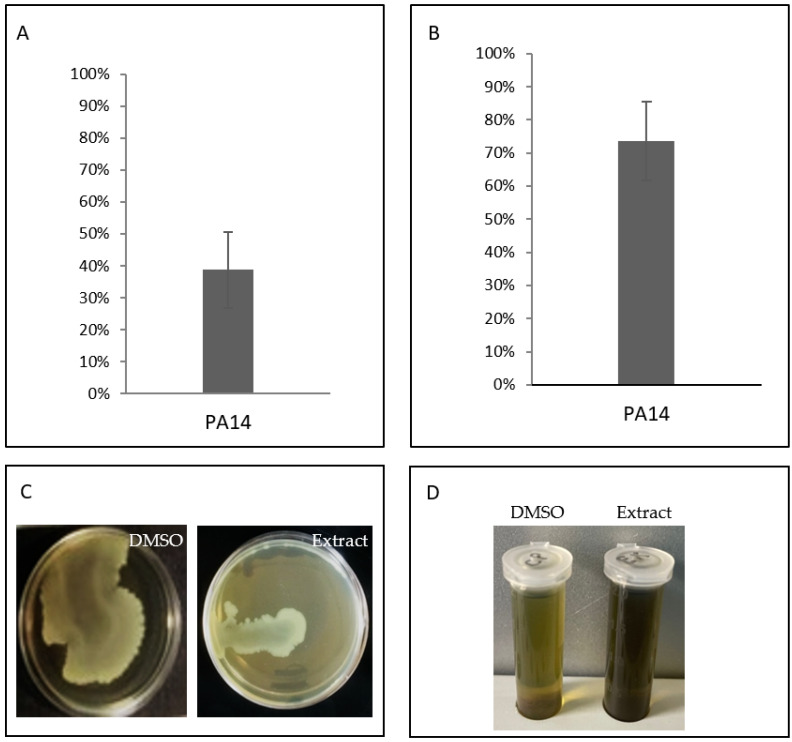
Effect of EX_TAE2020 on several virulence features of *P. aeruginosa* PA14. (**A)** Effect of EX_TAE2020 on biofilm formation of *P. aeruginosa* PA14. On the ordinate axis, the percentage of bacterial biofilm production is reported. Data are expressed as the percentage of biofilm formed in the presence of EX_TAE2020 compared with the untreated sample. Each data point is composed of 4 independent experiments, each performed in at least 3 replicates. (**B**) Effect of EX_TAE2020 on protease production of *P. aeruginosa* PA14 after 48 h of incubation. Data are reported as the percentage of residual proteolytic activity after the EX_TAE2020 treatment in comparison with that of untreated controls. Each data point is composed of 4 independent experiments, each performed in at least 3 replicates. (**C**) Swarming inhibition assay of EX_TAE2020 on *P. aeruginosa* PA14. (**D**) Effect of EX_TAE2020 on pyocyanin production of *P. aeruginosa* PA14 after 48 h of incubation.

**Figure 9 antibiotics-10-00944-f009:**
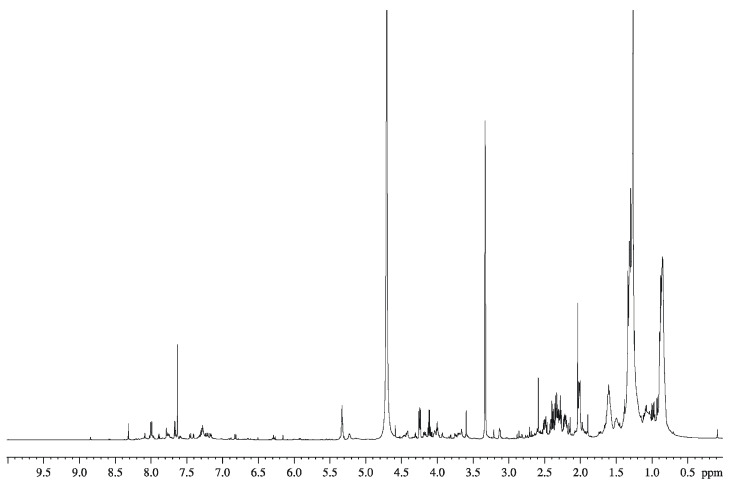
^1^H NMR spectrum of organic extract from SN_TAE2020 in CDCl_3_/CD_3_OD 1:1. The spectrum was recorded at 298 K, at 600 MHz.

**Table 1 antibiotics-10-00944-t001:** Phenotypic characterization of clinical and PA14 strains.

Bacterial Strain	Pyocyanin (OD 520 nm)	Protease Activity (OD 400 nm)	Biofilm 24 h ^a^ (OD 590 nm)	Biofilm 48 h ^b^ (OD 590 nm)	Swarming Motility ^c^	Swimming Motility ^c^
PA14	0.178 ± 0.042	2.037 ± 0.020	3.561 ± 0.357	13.470 ± 1.403	+++	+++
23P	0.148 ± 0.024	1.988 ± 0.057	3.175 ± 0.851	0.738 ± 0.373	+++	+++
27P	0.102 ± 0.005	2.175 ± 0.036	1.429 ± 0.643	3.049 ± 0.796	+++	+++
31P	0.063 ± 0.031	1.915 ± 0.001	1.741 ± 0.154	5.133 ± 0.946	+++	+++
32P	0.121 ± 0.037	2.439 ± 0.078	1.117 ± 0.163	5.597 ± 1.390	+++	+++
40P	0.160 ± 0.041	2.064 ± 0.058	0.970 ± 0.201	2.172 ± 0.194	+++	+++

^a^ Biofilm production during an incubation period of 24 h without medium replacement. ^b^ Biofilm production during an incubation period of 48 h with medium replacement after 24 h. ^c^ Motility zone (mm) < 15 mm; +++ motility zone (mm) > 30 mm.

**Table 2 antibiotics-10-00944-t002:** The five *P. aeruginosa* clinical isolates and their characterization by several properties [41].

ID pt	ID	SAMPLE	Date	Phenotype	1st Infection	Early Infection	Late Infection
22	23P	ESP	6/24/2005	sm		X	
24	27P	AT	1/31/2017	sm			X
9	31P	ESP	1/11/2017	M			X
26	32P	AT	12/5/2006	sm	X		
30	40P	AT	7/1/2013	i	X		

ID pt: patient identification; ID: strain code; Date: date of collection; Esp: sputum; AT: hypopharyngeal suction; TF: throat swabs; sm: smooth phenotype; i: irregular colony edges; m: mucoid colony; X: denotes positive for the feature.

## Data Availability

Not applicable.

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
