# Peer review of "Anti-Virulence Activity of the Cell-Free Supernatant of the Antarctic Bacterium Psychrobacter sp. TAE2020 against Pseudomonas aeruginosa Clinical Isolates from Cystic Fibrosis Patients"

_antibiotics, 2021, doi:10.3390/antibiotics10080944_

Round 1

Reviewer 1 Report

Dear Authors,

It is with great pleasure that I read this work on anti-virulence properties of supernatant of the bacterium Psychrobacter sp TAE2020. Although the work is not novel, overall it is indeed very interesting. I found that the experiments were well designed and thoroughly described in the text.

In my opinion these shortcomings must be addressed

  • The supernatant did not show antimicrobial activity at all tested dilutions, but I would like to see the growth curves, at least for the dilutions later mentioned in the paper like 1/2 and 1/5. Is it possible that these phenotypes are due to growth inhibition of the supernatant?
  • Along the similar line of thought, the protease and pyocyanin assays should be normalised to cell number (optical density)
  • Line 135 mentions the control but the results from this control condition is not described adequately
  • Grammar and spelling mistakes on lines 47-48, 86 and 272

In addition, I have the following suggestion to improve the article

  • Instead of using the phrase "biofilm aspect" biofilm topography can be used

Author Response

It is with great pleasure that I read this work on anti-virulence properties of supernatant of the bacterium Psychrobacter sp TAE2020. Although the work is not novel, overall it is indeed very interesting. I found that the experiments were well designed and thoroughly described in the text.

We thank the reviewer for these comments.

The supernatant did not show antimicrobial activity at all tested dilutions, but I would like to see the growth curves, at least for the dilutions later mentioned in the paper like 1/2 and 1/5. Is it possible that these phenotypes are due to growth inhibition of the supernatant?

We agree with the reviewer’s suggestion and the growth curves were added in the manuscript in the results section. Results obtained showed that the growth curves in the presence of SV_TAE2020 are nearly superimposable to the control curves. We also analyzed the growth behavior of the bacterial strains in the presence of 50% of the culture medium used for TAE2020 growth (Gut broth), as reported in the graphics represented in Figure 1.

The corresponding paragraph in Results section has been added (lines 144-152). Furthermore, in the Discussion section this point has been well defined (line 460).

Along the similar line of thought, the protease and pyocyanin assays should be normalized to cell number (optical density)

We agree with this comment and we done it.

Lines 217-218 and line 596: we added this information for pyocyanin production and the corresponding figure was emended (Figure 3).

Lines 230-231 ald lines 607-608: we added this information for proteolytic activity and the corresponding figure was emended (Figure 4).

Line 135 mentions the control but the results from this control condition is not described adequately

We agree with the reviewer’s comment and the manuscript has been correspondingly emended.

In line 169-170 we corrected the phrase as reported: “Results were expressed as the percentage of biofilm formed in presence of SN_TAE2020 compared to the bacteria grown in the presence of 50% of Gut medium”. 

In line 178 we added the following sentence: “As control, the growth medium supplemented with Gut broth at a dilution of 1:2 was added in the corresponding wells”.

We also corrected the legend of figure 1 (lines 199 and 202).

The corresponding paragraph in Materials and Methods section has been emended (lines 578-579).

Lines 215 and 230, we corrected the description of experiments taking account the performed controls.

We also corrected the legends of figure 3 (lines 235-236) and figure 4 (lines 249-250).

Grammar and spelling mistakes on lines 47-48, 86 and 272

We apologize for this spelling mistakes. We corrected them as reported below.

Lines 48-50 the previous phrase was substituted by: “P. aeruginosa is an aerobe and facultative anaerobe that can well survive in micro-aerobic conditions such as those generated by the sticky mucus within lungs of CF patients [5,6]”.

Line 88 “adaptat” in “adapt”

Line 327 “disgregated” in “disaggregated”

Instead of using the phrase "biofilm aspect" biofilm topography can be used

We thank the reviewer for this comment and the phrase biofilm topography was substituted in the text (lines 326 and 345)

Reviewer 2 Report

This is an interesting study and the data presented by the authors is very convicent.

I have minnor recommendations. The title and some sections of this manuscript need some adjustsments. 

Although is clear that cell-free supernatant of the Antarctic bacterium Psychrobacter sp TAE2020 inhibits the P aeruginosa biofilm formation. There is lack of information on its anti-virulent potential (e.g. the authors did not performed any transcriptomic analysis, more complex in vitro studies were not done including mixing P aeruginosa with mammalian cells nor the study has in vivo data). So, I think this study is in its initial steps, but I see valuable contribution of the study in future publications of the authors.

Author Response

This is an interesting study and the data presented by the authors is very convincing.

I have minor recommendations. The title and some sections of this manuscript need some adjustments. 

Although is clear that cell-free supernatant of the Antarctic bacterium Psychrobacter sp TAE2020 inhibits the P aeruginosa biofilm formation. There is lack of information on its anti-virulent potential (e.g. the authors did not performed any transcriptomic analysis, more complex in vitro studies were not done including mixing P aeruginosa with mammalian cells nor the study has in vivo data). So, I think this study is in its initial steps, but I see valuable contribution of the study in future publications of the authors.

We thank the reviewer for his/her suggestions.

We agree that an understanding at the molecular level of the mechanism of Psychrobacter sp TAE2020 supernatant or extract requires the analysis of the expression of many virulence factors (including proteases LasA and LasB, pili associated genes involved in swarming like the type IV secretion system, flagellar proteins, etc.). Furthermore, also the investigation of TAE2020 supernatant activity during the invasion of P. aeruginosa on eukaryotic cells can help to understand its mechanism of action.

This work is, as reviewer has underlined, an initial step because the future goals of the authors are the identification and the purification of these molecules with anti-virulence activity. Once these compounds will be identified, their anti-virulent potential will be certainly investigated at a transcriptomic level and their effect on eukaryotic cells during the P. aeruginosa infection will be also studied.

The aim of the present work was the investigation of some characteristics associated with the virulence of P. aeruginosa and we thanks to the reviewer for her/his stimuli. For this reason, the title of the manuscript has been modified substituting “anti-virulence properties” with “anti-virulence activity”.

Round 2

Reviewer 1 Report

Dear Authors,

I wish you good luck for future endeavors.